# Impact of Tillage and Fertilization on $CO_2$ Emission from Soil under Maize Cultivation

**Liliana Salinas-Alcántara, Rocio Vaca, Pedro del Águila, Nadia de la Portilla-López, Gustavo Yáñez-Ocampo, Laura A. Sánchez-Paz and Jorge A. Lugo ***

Laboratorio de Edafología y Ambiente, Facultad de Ciencias, Universidad Autónoma del Estado de México, Instituto Literario No. 100, Toluca 50000, Mexico; lsalinasa606@alumno.uaemex.mx (L.S.-A.); rvp@uaemex.mx (R.V.); daguila@uaemex.mx (P.d.Á.); ndelaportillal@uaemex.mx (N.d.l.P.-L.); gyanezo@uaemex.mx (G.Y.-O.); lasanchezp@uaemex.mx (L.A.S.-P.)
\* Correspondence: jlugo@uaemex.mx; Tel.: +52-7222965556-162

**Abstract:** Agriculture is in a constant state of change. Its new practices and technologies represent impacts that are difficult to predict. The transition from animal traction to tractors and the substitution of manure for synthetic fertilizers are changes that are taking place particularly in developed countries, yet they are increasing in developing ones. However, the effect of these changes on agriculture and soil $CO_2$ emissions remains controversial. The objective of this study was to measure the effects of two tillage systems and fertilization on the $CO_2$ emissions from the soil under maize cultivation. Therefore, it consisted of two tillage systems, namely tractor (T) and animal (A) traction, and four fertilization methods. The fertilization treatments tested were: (CH) application of N, P, K chemical fertilizer; (HM) application of horse manure; (CM) application of chicken manure; and (CT) unfertilized control. We found that the soil $CO_2$ emission rates in the maize growing season was higher than those in the tillage before the harvest season. Soil respiration peaked in June after the second fertilizer application (9394.59–12,851.35 mg $CO_2$ m$^{-2}$ h$^{-1}$ at tractor and 7091.89–12,655.86 mg $CO_2$ m$^{-2}$ h$^{-1}$ at animal traction). The production of corn grain only presented differences between the treatments with and without application of fertilizers.

**Keywords:** animal traction; tractor; manure; fertilizer; agriculture

## 1. Introduction

The increase in greenhouse gases (GHG) in the atmosphere is currently one of the greatest concerns to researchers. Agriculture is a sector that contributes to this increase, accounting for a third of global GHG [1]. Among the main agricultural practices that cause the release of these gases into the atmosphere are the production of fertilizers and pesticides, the increase in the use of machinery, the change in land use, and, consequently, soil degradation [2,3]. Agriculture emits approximately 500 Tg C per year, and it is predicted that by the year 2030 it will reach the equivalent to 8.3 Gt of $CO_2$ [4–6]. Therefore, the reduction in $CO_2$ emissions from agriculture represents a great challenge, due to its lasting effect on the atmosphere, which makes it urgent to reduce its balance by 2050 [7–9]. In particular, the application of nitrogenous fertilizers can generate between 187 and 224 Tg of $CH_4$ and between 1.7 and 4.8 Tg of $N_2O$ per year [10].

Agriculture has experienced various technological changes throughout its history, both in the improvement of seeds, the rise in the production and application of fertilizers and pesticides, and the implementation of machinery [11,12]. These changes in agricultural practices have repercussions both on society and the environment since they influence the emission of $CO_2$ from agricultural soil [13–15].

Tillage is one of the main agricultural practices in maize production and it is used to prepare the seedbed and optimize soil conditions. It aims to stimulate seed germination, development, and growth of seedlings [4,7]. Its effect depends on factors such as depth,

soil type, and the tillage method used [16]. However, inadequate tillage practices can exert a change in the physical and chemical properties of the soil, in turn causing the release of $CO_2$ [17]. During intensive tillage operations, soil aggregates are broken down and the organic matter (OM) contained within them is exposed, facilitating the oxidation of soil C in addition to releasing the $CO_2$ contained in the soil pores [18].

The $CO_2$ emission product of zero tillage is the main method evaluated. However, there are few studies on the impact of the different types of tillage on corn production [4]. This crop is one of the main cereals cultivated worldwide and presents differences in the emission of $CO_2$ from the soil with different tillage systems [19]. For years, draft animals have been representative of peasant agriculture. However, these are currently used especially in developing countries, in areas of difficult access, or in small extensions [20]. Recently, interest has been generated in the use of draft animals instead of tractors for soil tillage tasks due to the search for safe practices for the environment [21]. In Mexico, 93.5% of the production units have surfaces smaller than 20 ha. The use or introduction of mechanical traction is difficult when the extension of the production units is small and topographically irregular, and the use of animal and human traction is important in agricultural production [22].

Greenhouse gas emissions from tillage are defined by output efficiency, energy source, and gas waste makeup. The net efficiency of draft animals and tractors is similar, with both converting 30% of input energy into useful energy [23]. The difference lies in the fuels used for each tillage method, but in the end both emit GHG into the atmosphere [24].

The emission of $CO_2$ from agricultural soils increases after the application of nitrogenous fertilizers [25]. Nitrogen represents the main nutrient for crops, and therefore the application of mineral fertilizers has increased considerably (replacing organic fertilizers) [26]. On the other hand, the evaluation of GHG emissions related to the application of manure is complex and varies according to its composition and its management conditions [27]. The increase in $CO_2$ emission due to the addition of fertilizers can be attributed to the respiration of the roots, the greater activity of the microorganisms, and the decomposition of the OM [28]. In addition, soil respiration can present variations in the different seasons of the crop. Some authors report the highest flow of $CO_2$ in the plant growth season, but they also highlight the importance of the emission of this gas in the winter season or senescence of the crop [1,29–31].

Tillage and the application of mineral fertilizers require a lot of energy compared to draft animals and the use of manure, and they are also directly or indirectly related to pollution [32,33]. Therefore, alternative agricultural practices that help reduce GHG emissions and maintain soil fertility should be sought and evaluated [3–6].

The objective of this study was to evaluate the effects of two tillage systems and fertilization on soil $CO_2$ emission in maize crops. The specific objectives were to: (1) understand the implications for tractors compared to animals for traction, and manure compared to mineral fertilizers; (2) evaluate the seasonal emission of $CO_2$; and (3) compare the resulting corn production.

## 2. Materials and Methods

### 2.1. Site Descriptions

The experiment was carried out in Jiquipilco (19°42′N, 99°40′W and 2550 m above mean sea level), located in the North of the State of Mexico, Mexico (Figure 1). The climate of the region is characterized by tropical, rainy summers, and dry winters [34], with an average annual rainfall of 881.7 mm and mean yearly temperature of 13.2 °C. The soil is planosol eutric (We) and vertisol pelic (Vp). This clay soil does not have good drainage, which favors the flooding of the land when there is an excess of water, making it difficult to implement conservation practices such as zero tillage. In addition, in this type of soil, agricultural activities can be developed with an optimal use of the crop (one of the main ones is corn (*Zea mays* L.) in this region). This is an annual crop, mainly cultivated during spring and summer [35]. The study area is located in a rural production area (for self-

consumption), where the available sources of fertilizers are manure and chemical fertilizers. The study on $CO_2$ emission from the soil was carried out in 2020–2021. It was carried out under the conditions of field experiments. Two maize cultivation systems were applied, namely tillage with tractor and animal traction. In both cultivation systems, the planting depth was 20 cm and the space between rows was 75 cm. The corn was planted during the first week of April and harvested in mid-October. The study was carried out under conditions depending on the weather. This study reports data on the $CO_2$ emission of the tillage and fertilization season and subsequent management to present the complete cycle of the corn crop. Additionally, traditional methods were used to work the field throughout the duration of the experiment. These involved the main methods of fertilization, manual harvesting, weed control by hand, and herbicides (Table 1).

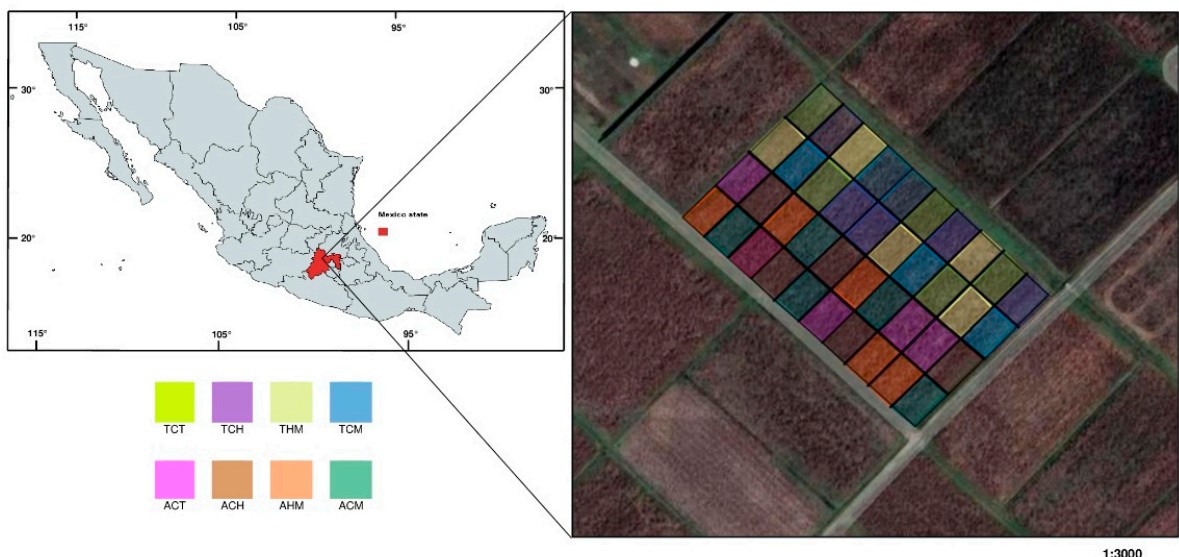

**Figure 1.** Location map of the experimental area. TCT, tractor unfertilized control; TCH, tractor chemical fertilizer; THM, tractor horse manure; TCM, tractor chicken manure; ACT, animal unfertilized control; ACH, animal chemical fertilizer; AHM, animal horse manure; ACM, animal chicken manure.

**Table 1.** Management practices for corn cultivation during 2020.

| Farming Operation | Date |
|---|---|
| First tillage | 10 Mar |
| Maize sowing | 2 April |
| First fertilization | 20 May |
| Second tillage | 7 June |
| Second fertilization | 14 July |
| Herbicide application | 1 August |
| Harvest | 10 November |
| Plant cut | 5 December |

*2.2. Experimental Treatments*

In this experiment, a multifactorial design was carried out, with eight treatments and five repetitions. Five replicates were made for each treatment for a total of 40 experimental plots. The size of each plot was 10 m by 10 m (100 m$^2$). The experiment consisted of two tillage systems: tractor (T) and animal (A) traction, and four fertilization methods. The fertilization treatments tested were (CH) use of N, P, K chemical fertilizer (180 kg·ha$^{-1}$); (HM) use of horse manure (200 kg·ha$^{-1}$); (CM) application of chicken manure (135 kg·ha$^{-1}$); and (CT) an unfertilized control.

### 2.3. Soil $CO_2$ Emission Measurements

The rate of $CO_2$ emission from the soil was measured by a closed-chamber system. The emission was measured daily six days after each farming operation. We used PVC (polyvinyl chloride) columns with a cap (diameter 16.1 cm) as a closed-chamber [36]. A $CO_2$ trap was prepared using NaOH solution (0.1 N) in three vials for trapping $CO_2$, and these vials were placed inside of each chamber. PVC columns were inserted randomly in each plot (five PVC columns per site) at a depth of 2 cm into the soil in order to avoid soil disturbance and the associated undesirable emissions. The vials with alkali solution were removed and titrated with a 0.1 N HCl solution, using $BaCl_2$ and phenolphthalein indicator solutions. As controls of the experiment, we placed the vials with NaOH inside hermetic glasses [37]. All field measurements were conducted between 12:00 and 15:00 h.

The $CO_2$ emission from the soil during exposure to alkali was calculated with the following formula:

$$\text{Milligrams of } CO_2 = (T - C)\,(N)\,(E)\,(Vtr/Vti)$$

where T is the volume (mL) of acid needed to titrate NaOH in the containers from the control; C is the volume (mL) of acid needed to titrate the NaOH in the containers exposed to the soil atmosphere; N is the normality of the acid. To express the data in terms of carbon, E = 6; to express it as $CO_2$, E = 22. Vtr is the volume (mL) of NaOH for each jar. Vti is the volume (mL) of NaOH used to titrate. The daily emission of carbon dioxide was expressed as mg $CO_2$ day$^{-1}$ ha$^{-1}$ soil [38].

### 2.4. Soil Sampling Analyses

Soil samples were collected after the first tillage; the first 20 cm were taken for each treatment and its repetitions (40 samples). The soil samples were kept separately in plastic bags and transported to the laboratory. The soil samples were dried and air sieved (<2 mm). Soil pH was determined in water 1:2 by the AS-02 method. Soil textures were determined according to Bouyoucos AS-09, electrical conductivity (EC) of the AS-18 method and organic matter (OM) of the AS-07 method [39].

### 2.5. Corn Cob Sampling

Ear samples were taken for each treatment. For this purpose, 2 linear rows of 10 m were harvested with a separation of 75 cm. The ears were counted and air-dried in the shade until their state of humidity allowed them to be manually shelled. Subsequently, the samples with and without cob were weighed in order to evaluate the effect of the tillage and fertilization system on grain yield.

### 2.6. Data Analysis

Repeated MANOVA measurements were performed to examine the effects of tillage and fertilization on $CO_2$ emission and soil properties, as well as a Tukey test to identify differences. All analyses were conducted using Statgraphics Centurion version XVl at a confidence level of 95%. The difference in results was considered statistically significant when the confidence interval was greater than 5% with ($p < 0.05$).

## 3. Results

### 3.1. Soil Properties

The physical and chemical properties of the soil were based on NOM-021-RECNAT-2001. The soil presented a predominantly clay loam texture. All treatments showed an apparent density from 1.03 to 1.22 g/cm$^3$ without differences. The soils of all sites were slightly acidic and had negligible salinity effects (Table 2).

**Table 2.** Properties of the soils in field experiment from all treatments.

| Treatments | Texture | BD (g cm$^3$) | pH | EC (dsm$^{-1}$) | OM (%) |
|---|---|---|---|---|---|
| TCT | | $1.17 \pm 0.10$ a | $5.24 \pm 0.24$ a | $0.21 \pm 0.01$ a | $1.59 \pm 0.15$ a |
| TCH | Clay loam | $1.08 \pm 0.60$ a | $5.02 \pm 0.23$ a | $0.27 \pm 0.03$ ab | $1.43 \pm 0.19$ a |
| THM | | $1.22 \pm 0.07$ a | $5.14 \pm 0.11$ a | $0.41 \pm 0.05$ c | $1.70 \pm 0.10$ a |
| TCM | | $1.22 \pm 0.07$ a | $5.06 \pm 0.11$ a | $0.30 \pm 0.02$ b | $1.69 \pm 0.19$ a |
| ACT | | $1.13 \pm 0.04$ a | $5.28 \pm 0.30$ a | $0.24 \pm 0.01$ ab | $1.54 \pm 0.15$ a |
| ACH | Clay loam | $1.03 \pm 0.04$ a | $5.05 \pm 0.08$ a | $0.29 \pm 0.05$ ab | $1.70 \pm 0.10$ a |
| AHM | | $1.04 \pm 0.18$ a | $5.01 \pm 0.18$ a | $0.31 \pm 0.05$ b | $1.69 \pm 0.19$ a |
| ACM | | $1.08 \pm 0.07$ a | $5.02 \pm 0.07$ a | $0.28 \pm 0.01$ ab | $1.64 \pm 0.12$ a |

Average $\pm$ standard deviation. Different letters in the same column denote significant differences based on Tukey's test at $p < 0.05$. BD, bulk density; EC, electrical conductivity; OM, organic matter; TCT, tractor unfertilized control; TCH, tractor chemical fertilizer; THM, tractor horse manure; TCM, tractor chicken manure; ACT, animal unfertilized control; ACH, animal chemical fertilizer; AHM, animal horse manure; ACM, animal chicken manure.

*3.2. Soil CO$_2$ Emission*

Differences in soil CO$_2$ emission due to tillage treatment and different fertilization were analyzed for each of the managements of crop maize. The interaction between tillage and fertilizer had a significant effect on CO$_2$ efflux from the soil. Two-way ANOVA showed that fertilization has a marked influence on the emission of CO$_2$ from the soil ($p < 0.001$). And soil CO$_2$ emission was affected by the interaction of tillage $\times$ fertilization only after 2nd fertilizer ($p < 0.001$). Tillage alone did not show a significant effect on CO$_2$ emission (Table 3).

**Table 3.** Summary of multivariate analysis of variance for the effect of tillage and fertilization on the emission of CO$_2$ in different corn crop operations.

| | 1st Tillage | Sowing | 1st Fertilizer | 2nd Tillage | 2nd Fertilizer | Herbicide | Harvest | Plant Cut |
|---|---|---|---|---|---|---|---|---|
| Tillage (T) | 0.53 | 0.41 | 2.80 | 0.62 | 13.57 ** | 39.78 ** | 2.69 | 39.68 ** |
| Fertilization (F) | 1.80 | 2.64 | 8.19 ** | 33.67 ** | 145.65 ** | 5.70 * | 3.80 * | 6.10 ** |
| T $\times$ F | 0.08 | 0.49 | 0.53 | 1.85 | 5.31 * | 1.46 | 0.37 | 1.54 |

Values of F. * $p < 0.05$, ** $p < 0.01$.

The evolution of CO$_2$ output from the soil during the studied maize crop cycle showed peaks related to crop management, and they were particularly clear during the 2nd fertilizer management, and decreased after the flowering period. An increase in CO$_2$ emissions was observed, coinciding with the period of maximum maize growth and maximum rate of maize dry-matter accumulation. Soil respiration peaked in June after the second fertilizer (9394.59–12,851.35 mg CO$_2$ m$^{-2}$ h$^{-1}$ for tractor and 7091.89–12,655.86 mg CO$_2$ m$^{-2}$ h$^{-1}$ for animal traction). Fertilization resulted in a significant increase in CO$_2$ emission from the soil, where TCM (12,851.35 CO$_2$ m$^{-2}$ h$^{-1)}$) and ACM (12,655.86 mg CO$_2$ m$^{-2}$ h$^{-1)}$) were the treatments with the highest values of the whole maize crop cycle. After the increase in CO$_2$ emissions, these decreased progressively until the maize plant was cut (3945.59–4094.59 mg CO$_2$ m$^{-2}$ h$^{-1}$ for tractor tillage, and 3718.92–3520.72 mg CO$_2$ m$^{-2}$ h$^{-1}$ for animal traction) when they reached values lower than those obtained in the first tillage (Figure 2).

Table 4 shows the managements where there were differences in the soil CO$_2$ emission between the different treatments. For the first and second fertilizer applications, the average CO$_2$ output from the soil was significantly higher for the TCH (6021.62 CO$_2$ m$^{-2}$ h$^{-1}$), TCM (6605.41 CO$_2$ m$^{-2}$ h$^{-1}$), ACM (6248.65 CO$_2$ m$^{-2}$ h$^{-1}$) and TCM (12,702.70 CO$_2$ m$^{-2}$ h$^{-1}$), ACM (12,655.90 CO$_2$ m$^{-2}$ h$^{-1}$) treatments, respectively. On the other hand, during the second tillage period, TCH and ACM were the treatments with the highest values. In the control treatments, higher soil CO$_2$ emissions were observed under tractor tillage compared to animal traction tillage.

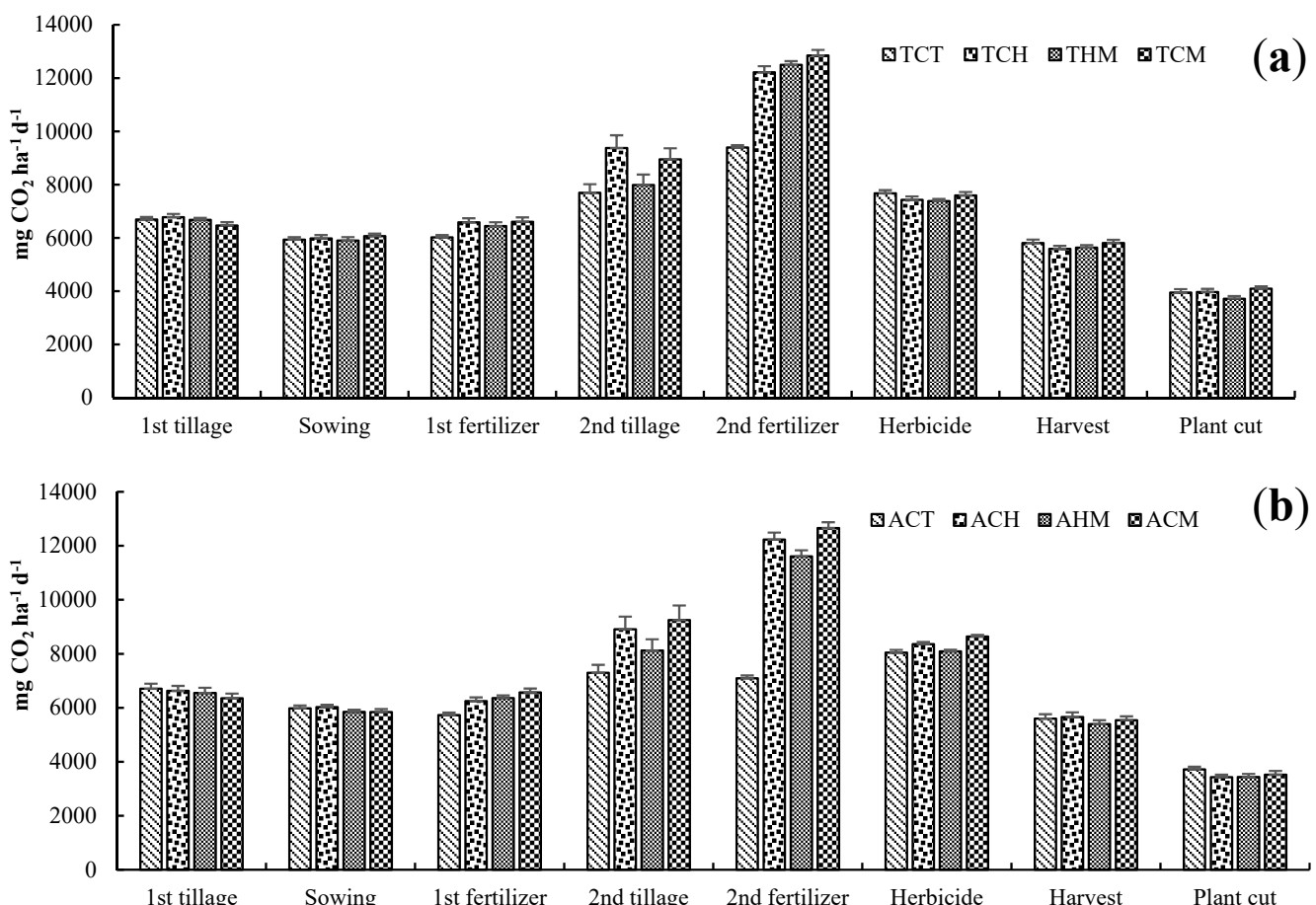

**Figure 2.** Mean soil $CO_2$ emission rate during different crop growth periods under different tillage and nitrogen fertilizer treatments. (**a**) Tillage tractor; (**b**) animal traction. TCT, tractor unfertilized control; TCH, tractor chemical fertilizer; THM, tractor horse manure; TCM, tractor chicken manure; ACT, animal unfertilized control; ACH, animal chemical fertilizer; AHM, animal horse manure; ACM, animal chicken manure. The error bars represent the sample population variation.

**Table 4.** Shows the significant differences between the treatments ($CO_2$ m$^{-2}$ h$^{-1}$).

| Treatments | 1st Fertilizer | 2nd Tillage | 2nd Fertilizer |
|---|---|---|---|
| TCT | 6021.62 ± 302.41 ab | 7697.30 ± 211.43 c | 8821.62 ± 454.50 c |
| TCH | 6583.79 ± 313.79 a | 9372.97 ± 456.43 a | 12,205.40 ± 233.75 ab |
| THM | 6227.03 ± 405.22 ab | 7989.19 ± 550.45 bc | 12,097.30 ± 922.41 ab |
| TCM | 6605.41 ± 149.99 a | 8951.35 ± 418.00 ab | 12,702.70 ± 380.31 a |
| ACT | 5729.73 ± 275.62 b | 7297.30 ± 354.45 c | 7091.89 ± 196.39 d |
| ACH | 6569.37 ± 140.95 ab | 8908.11 ± 320.70 ab | 12,227.00 ± 593.86 ab |
| AHM | 6172.97 ± 403.42 ab | 8118.92 ± 549.12 c | 11,221.60 ± 745.67 b |
| ACM | 6248.65 ± 533.53 a | 9246.85 ± 541.95 a | 12,655.90 ± 588.64 a |

Average ± standard deviation. Different letters in the same column denote significant differences based on Tukey's test at $p < 0.05$. TCT, tractor unfertilized control; TCH, tractor chemical fertilizer; THM, tractor horse manure; TCM, tractor chicken manure; ACT, animal unfertilized control; ACH, animal chemical fertilizer; AHM, animal horse manure; ACM, animal chicken manure.

Finally, $CO_2$ emissions accumulated in the soil were higher in tractor tillage compared to animal traction tillage. In addition, the emissions were significantly higher for TCH, TCM and ACM ($p < 0.05$), while THM and AHM were the treatments with the lowest $CO_2$ emissions after the control treatments (Table 5).

**Table 5.** Accumulated soil $CO_2$ emissions for all treatments during the maize cropping period, for the 8 managements (average $\pm$ SD).

| Treatments | Accumulated Soil ($CO_2$ $m^{-2}$ $h^{-1}$) |
|---|---|
| TCT | $263,243.24 \pm 906.11$ cd |
| TCH | $290,324.32 \pm 958.91$ a |
| THM | $272,810.81 \pm 2818.24$ bc |
| TCM | $290,342.34 \pm 662.44$ a |
| ACT | $250,810.81 \pm 831.27$ d |
| ACH | $287,351.35 \pm 885.23$ ab |
| AHM | $268,108.11 \pm 2860.27$ c |
| ACM | $291,819.82 \pm 1248.74$ a |

Different letters denote significant differences based on Tukey's test at $p < 0.05$. TCT, tractor unfertilized control; TCH, tractor chemical fertilizer; THM, tractor horse manure; TCM, tractor chicken manure; ACT, animal unfertilized control; ACH, animal chemical fertilizer; AHM, animal horse manure; ACM, animal chicken manure.

### 3.3. Corn Production

The highest number of ears per hectare was presented by ACM ($42,000.00 \pm 4807.40$), ACH ($42,400.00 \pm 11,240.8$) and TCH ($37,733.30 \pm 7595.32$). A similar trend is observed in the weight of the cob with cob per hectare, where the same treatments have the highest weight and ACT ($2.29 \pm 0.25$ ton/ha) is the treatment with the lowest weight (Table 6). However, the weight of the corn grain without cob did not present any significant differences in the treatments with some types of fertilization, with TCT and ACT being the treatments that showed the lowest yield.

**Table 6.** Average corn production of each treatment.

| Treatments | Number of Ears | Weight with Cob (ton/ha) | Weight without Cob (ton/ha) |
|---|---|---|---|
| TCT | $30,000.00 \pm 5142.42$ cd | $3.11 \pm 0.67$ bc | $2.57 \pm 0.41$ b |
| TCH | $37,733.30 \pm 7595.32$ ab | $4.40 \pm 0.91$ a | $3.95 \pm 0.81$ a |
| THM | $35,866.70 \pm 2641.55$ bc | $3.84 \pm 0.35$ ab | $3.45 \pm 0.34$ a |
| TCM | $33,333.30 \pm 3972.12$ bc | $3.86 \pm 0.80$ ab | $3.55 \pm 0.75$ a |
| ACT | $27,600.00 \pm 1211.06$ d | $2.29 \pm 0.25$ c | $2.26 \pm 0.37$ b |
| ACH | $42,400.00 \pm 11,240.8$ a | $4.41 \pm 0.88$ a | $3.93 \pm 0.83$ a |
| AHM | $39,466.07 \pm 3870.12$ ab | $3.86 \pm 0.55$ ab | $3.47 \pm 0.49$ a |
| ACM | $42,000.00 \pm 4807.40$ a | $4.36 \pm 0.61$ a | $3.90 \pm 0.55$ a |

Average $\pm$ standard deviation. Different letters in the same column denote significant differences based on Tukey's test at $p < 0.05$. TCT, tractor unfertilized control; TCH, tractor chemical fertilizer; THM, tractor horse manure; TCM, tractor chicken manure; ACT, animal unfertilized control; ACH, animal chemical fertilizer; AHM, animal horse manure; ACM, animal chicken manure.

## 4. Discussion

### 4.1. Tillage and Fertilization Effects on $CO_2$ Emissions

Agriculture is in a constant change of practices and new technologies, which represent impacts that are difficult to predict. The transition from oxen to tractors, and the consequent substitution of manure for mineral fertilizers, are changes that are occurring in agriculture in developing countries [13,20]. This study showed the effect of two tillage systems and different fertilization treatments on $CO_2$ emission throughout a complete maize crop cycle. During the maize growing season, agricultural practices and crop growth greatly affected $CO_2$ output from the soil, where the effect of tillage and fertilization on $CO_2$ emissions was evident from the second dose of fertilization (Table 3). This concurs with what was reported by Salamanca-Fresno et al. [40], who found a greater effect of agricultural practices on environmental conditions. Our results showed a marked effect of fertilization and tillage, just after the second fertilizer application. However, there is still no agreement on the effect of Nitrogen availability on soil C mineralization, with works showing a stimulating or suppressive effect [41,42]. It is also important to note that the effect of tillage and fertilization on subsequent management, such as the application of herbamine, harvesting

and cutting of corn, is a factor to be taken into account in the estimation of global $CO_2$ emissions from the cultivation of this grain.

### 4.2. Temporal Variation of $CO_2$ Emissions

Our results present higher soil $CO_2$ fluxes than other works in corn cultivation [3,43,44], but are similar to the results obtained by Salamanca-Fresno et al. [40], in which measurements were also taken within plant rows and in between rows. The highest peaks of $CO_2$ emission occurred after the second tillage, second fertilization and weed control (Figure 2). This period is known to coincide with the rainy season. As mentioned above, the site where the experiment was carried out is characterized by rains in summer. Therefore, neither the temperature nor the humidity of the soil were limiting factors, resulting in the growth of crops and greater soil respiration [45]. Jans et al. [46] mention that the highest soil $CO_2$ emissions take place during the growth phase of the corn plant and may be associated with roots and soil respiration, the decomposition rate of soil OM, and crop residues. Soil $CO_2$ emissions are highest during the early vegetative growth phases and then decline in reproductive and senescence phases. Then, it is subject to phenological development and on C distribution within plants [3,19,47]. On the other hand, tillage causes the breakdown of soil aggregates and exposes OM to be degraded by microorganisms, thus increasing microbial activity, in addition to releasing $CO_2$ from soil pores [18].

In addition, when the second dose of fertilizer was added, both tillage systems reacted with a rapid increase in $CO_2$ output from the soil (Figure 2). For the TCH and ACH treatments, this may be the result of the application of nitrogenous fertilizers that can promote C reserves especially by increasing root biomass, which may contribute to a more stable SOC than aboveground residues. Also, the long-term application of organic and inorganic fertilizers improves the soil organic carbon. On the other hand, the increase in $CO_2$ emissions from the THM, TCM, AHM, and ACM treatments could be explained by the addition of organic fertilizers, which, in addition to maintaining or improving crop yields and SOC reserves, also have significant effects in $CO_2$ emissions [48].

The differences appeared after the first application of fertilizers (Table 4), similar to the data of Li et al. [1], where the contribution of tillage to accumulated soil $CO_2$ emissions was small since the fallow period was during the months with the lowest temperature and less precipitation. While TMC and ACM treatments showed the highest values, agricultural $CO_2$ emissions increase with fertilization [49,50]. Organic fertilizers supply sources of C for microbial activity, and therefore stimulate biochemical processes and root metabolism to improve the intensity of soil respiration [51]. However, nitrogen fertilization can have variable effects on $CO_2$ emissions [52]. The effect of fertilizers on soil $CO_2$ remains debatable [2].

After the application of the herbicide, the emission of $CO_2$ began to decrease in all treatments, which can be explained with the seasonal variation of $CO_2$, which tends to be higher in summer and decrease in winter [53]. However, the contribution of winter carbon dioxide must be considered when evaluating the annual carbon budget on a regional and global scale [54]. An important factor is the interaction between environmental variables and $CO_2$ emission. However, at a regional scale, climatic effects can be masked in the topsoil by land use/management, particularly on farmland soils, where intensive management (fertilization, irrigation, etc.) can counteract climatic effects [55]. Finally, the treatments that presented the highest $CO_2$ emission throughout the corn crop cycle were TCH, TCM, and ACM. The addition of manure promotes higher carbon storage than the application of mineral fertilizer [56,57]. Also, organic fertilizer contains different nutrients that promote crop development and increase the soil organic carbon stock [58].

It is relevant to evaluate the role of the tractor and animal traction in the emission of $CO_2$ from agricultural soils, since the latter is not only used in developing countries, but also in countries such as the US and Germany, which use draft horses in numerous farms. However, studies on the effect of animal traction in agricultural systems are scarce. There

are several other reasons to resort to animal traction in the future if we consider the possible energy and environmental problems where a scarcity of fossil fuels may occur [20,59].

### 4.3. Corn Production

Fertilization has been a necessary practice to maintain soil fertility and improve crop productivity [60,61]. For this study, ACH, AMC and TCH were the treatments with the highest number of ears and the highest weight of the ear with cob; however, for the weight of the grain without cob, they only showed differences between the treatments with and without fertilization (TCT and ACT). In relation to grain corn, it is stated that "the yield in Mexico reaches an average of 3.2 tons/ha, with a temporary yield of 2.2 tons/ha" [62]. The results show that fertilization with manure is a viable option to maintain grain production (specifically horse manure, which presents lower $CO_2$ emissions compared to chicken manure and chemicals).

## 5. Conclusions

This study demonstrates that the tillage system and the application of fertilizers can affect $CO_2$ emissions from the soil. Different types of fertilization under tractor tillage and animal traction tillage led to different $CO_2$ emission rates. The highest $CO_2$ emissions were observed after fertilization, which took place during the growing season of the maize plant and decreased with subsequent management. For both types of tillage, the treatments with the highest $CO_2$ emissions were those fertilized with chicken manure. Regarding the production of corn grain, a difference was found in relation to the control treatments. Corn grain production from manure fertilization treatments kept up with chemical fertilization, proving to be a viable option to maintain production.

**Author Contributions:** L.S.-A.: designed the experiments and wrote the original draft preparation. R.V.: formal statistical analysis. P.d.Á. and L.A.S.-P.: reviewed the writing of the article. G.Y.-O. and N.d.l.P.-L.: data curation. J.A.L.: Reviewed the results and the English wording of the article and made corrections. All authors have read and agreed to the published version of the manuscript.

**Funding:** This research was funded by Universidad Autónoma del Estado de México, grant number 4635/2019SF.

**Institutional Review Board Statement:** Not applicable.

**Informed Consent Statement:** Not applicable.

**Data Availability Statement:** Not applicable.

**Acknowledgments:** The Consejo Nacional de Ciencia y Tecnología (CONACyT) for the student grant.

**Conflicts of Interest:** The authors declare no conflict of interest.

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
