# Peer review of "Impact of Tillage and Fertilization on CO2 Emission from Soil under Maize Cultivation"

_agriculture, doi:10.3390/agriculture12040555_

Round 1

Reviewer 1 Report

Review for the manuscript titled „IMPACT OF TILLAGE AND FERTILIZATION ON CO2 EMISSION FROM SOIL UNDER MAIZE CULTIVANION by Salinas-Alcántara et al.

The manuscript presents a study regarding the influence of agricultural practices on CO2 emission in order to provide a good input on future management and options to maintain a high-quality production. In addition, such studies can be very useful for highlighting the influence of agricultural practices in climate change. After reading the present manuscript thoroughly I appreciate the content as valuable but it does not seem to be ready for publication and it needs major revision.

The main issues that need to be reconsidered are:

  • An overall English improvement is necessary (especially in Discussion chapter). The writing needs to be improved. Some sentences, were not properly presented.

  • The Material and methods chapter is necessary to be improved (see the notes from specific issues)

Specific issues through the manuscript:

Title

The title sounds very good.

Abstract

Is short and essential.

Introduction

It is well structured

Material and methods

Site description

Page 2, line 94 – Zea mays must be italicized

Page 2, line 95 – “in 2020 -2021” – space before “2021”

Page 3, line 98 – last week of what month?

I suggest to re-write the sub-chapter “Site description” offering a more precise data about the crops in relation with the soil type, cultivation systems, fertilizer application (when, where, type, etc.). A map of the studied area with all these details will offer a good insight of the experimental part, so I suggest to add it to this chapter.

Experimental treatments

I suggest to better explain the application of fertilization treatments: how many plots were considered, what kind of treatment was applied for each plot, the temporal distance between fertilizers application, etc.

Soil CO2 emission measurements

Explain why did you choose the 2 cm depth for inserted the PVC columns and if these measurements are taking into consideration only the CO2 emitted by the soil without any other interaction (i.e – plant-root respiration, microbial respiration, exchange with the atmosphere, etc.)

Explain why did you choose to carry out the measurements at six days after each harvest period and only in that period?

And which was the timespan for sample the gases?

Have you measured the soil moisture, bulk density or C content? If not, please explain why did you choose not to measure the parameters which can influence the CO2 emission? If so, please explain the analytical methods for each of these parameters

Add a citation for the Mg CO2 calculation formula.

Soil sampling analyses

Add the number of soil samples collected and analyzed, mentioning also the depth of sampling. You can add the soil sampling location on the map.

Corn cob sampling

Add a phrase to explain the reason for corn cob sampling.

Results

Soil properties

I suggest to better explain the obtain results and to correlate these results when possible. Add “(Table 1)” at the end of this chapter to link the text with the table.

Discussion

Page 7, line 262 – Correct “variation”

Page 7, line 275 – delete “on the other hand”

I suggest to add a more detailed discussion about the relation between CO2 emissions and soil properties.

Conclusions

Conclusions are a brief summary of your main results reflecting the importance of such studies.

Author Response

The text in red color is added to the manuscript.

An English language and style edition was made

Reviewer 2 Report

Review MDPI Agriculture

Paper Description

This is an interesting paper and it applies to agriculture methodologies that are important for CO2 impacts. What comes out in the conclusions is about CO2 emission and agriculture methodologies impacts and relationships.

 It is worth mentioning to the authors that CO2 efflux is not only dependent on soil properties and morphology but also on environmental parameters. It is known for example that under high pressure systems (clear skies) the efflux coming out is lower than when the same soil under similar thermodynamic conditions (RH slight changes) is under the influences of a cyclones. As indicated in Kim et al, (2017) changes in CO2 fluxes during the snow-covered period can be as much as 35% on the average. These results are significant, as wintertime CO2 emissions represent ~20% of annual soil-originated emissions. Although these variations are smaller than the ones capture in the presented experiment it is worth noticing that the background emission is also important.

Kim Y., Y. Kodama and G.J. Fochesatto (2017). “Environmental factors regulating winter CO2 flux in snow-covered black forest soil of Interior Alaska”. Geochemical Journal, Vol. 51, No. 4, P. 359-371, 2017

Review Summary: If meteorological conditions are not available then I would suggest the authors to discuss or provide a paragraph indicating what background level emission you have and what is its variability. The mentioned paper is one example.

Minor corrections

Line 124 change “whit” by with Line 148: change “for” by “to”

Line 151: change “P” by p . This is the normal way in statistics to symbolize the probability.

Figure 1: errors bars need to be explained. Is this the sample population variation? Or is this associated to instrumental/method errors?

Line 262: correct variin

Line 275: repeated expressions. Get rid of: “On the other hand,”

Author Response

(The authors gave the same response as above.)

Round 2

Reviewer 1 Report

The main issue that need to be reconsidered is:

The map has to be improved in order to offer the information regarding the location of the studied area with the position within the country, province or locality and then the detail with the studied plots. The colors from the image have to be explained, the scale and the compass have to be added.

Author Response

comments for reviewers

Reviewer 2 Report

The author's addressed my comments satisfactorily. 

Author Response

comments for reviewers